# The Boom in Nanomaterials for Built Heritage Conservation: Why Does Size Matter?

**DOI:** 10.3390/ma16083277

**Published:** 2023-04-21

**Authors:** Jorge Otero, Giovanni Borsoi, Luis Monasterio-Guillot

**Affiliations:** 1Department of Mineralogy and Petrology, University of Granada, 18071 Granada, Spain; 2CERIS, Technical University of Lisbon, 1049-001 Lisbon, Portugal; 3ISTerre, University of Grenoble Alpes, 38000 Grenoble, France

**Keywords:** nanolime, nanomaterials, cultural heritage conservation, nanoscience, nanotechnology

## Abstract

There is no doubt that nanotechnology and nanoscience open new doors to new applications and products that can potentially revolutionize the practice field and how we conserve built heritage materials. However, we are living at the beginning of this era and the potential benefits of nanotechnology to specific conservation practice needs are not always fully understood. This opinion/review paper aims to present reflections and answer a question that we are often asked when working directly with stone field conservators: why should we use a nanomaterial instead of a conventional product? Why does size matter? To answer this question, we revise the basic concepts of nanoscience with implications for the built heritage conservation field.

The concept of nanoscience and/or nanotechnology was originally introduced in 1959 by the Nobel Prize Laureate in Physics, Richard Feynman, at the annual American Physical Society meeting at Caltech (USA), in his acclaimed lecture “There’s Plenty of Room at the Bottom” [1], where he hypothesized “*Why can’t we write the entire 24 volumes of the Encyclopedia Britannica on the head of a pin*?”. In this lecture, he introduced, for the first time, ideas about the possibility of direct manipulation of individual atoms and molecules to “*arrange them the way we want*”, and carry out intentional chemical synthesis by mechanical manipulation. Although he did not use the term “*nanotechnology*”, the ideas of this visionary lecture inspired the majority of scientists during the boom in nanoscience and nanotechnology at the beginning of the 1980s [2]. The definition of nanoscience is as simple as the phenomena that occur in systems at the nanometer scale. It essentially deals with the understanding, manipulation, and control of matter at dimensions between approximately 1 and 100 nanometers [3]. 

The use of nanoscience and nanotechnology has been around since ancient times. Since the time of the Greeks, approximately 2500 years ago (e.g., Democritus, [4]), nanoscience has been present in many applications of human technology or society, through questions about the infinitely fractioning of matter, nowadays called “*atoms*”. Experiments to modify matter up to the nanoscale level played a significant role in humankind’s development and this is clearly visible throughout history when we examine historic artefacts. One of the most famous examples is the Lycurgus Cup (4th-century AD Roman glass, dedicated to the death of King Lycurgus) at the British Museum [5], which appears green in daylight (reflected light), but red when light is transmitted from the inside of the vessel (Figure 1a). This effect was intentionally achieved by adding tiny proportions of nanoparticles of gold and silver dispersed in colloidal form throughout the glass material during the manufacturing process [6]. A similar example was found in medieval Lustre ceramic decorations [7], which are characterized by their unique metallic shine and colored iridescence on the surface [8]. This effect was intentionally achieved during the manufacturing process, where a nanocrystal film was created and responsible for the optical behavior and appearance of the decoration [9]. Another significant example could be the historical use of aqua regia to create the pigment Purple of Cassius since the Middle Ages [10]. When mixing aqua regia with gold, aqua regia converts metallic gold into gold ions, forming auric chloride complexes that, exposed to an aqueous solution of tin chloride (SnCl_2_), produce a precipitate of gold nanoparticles with an approximate size of 70 nanometers, which gives them a purple color instead of the characteristic yellow appearance [11]. This reaction can even occur spontaneously in nature under very specific scenarios, as recently evidenced in corroded gilded tin plasterwork in the Alhambra palaces (Granada, Spain) [12]. Those are just three examples of matter alteration at the nanoscale throughout history. However, modern nanotechnology truly began in 1981, when the scanning tunnelling microscope (STM) allowed scientists and engineers for the first time to see and manipulate individual atoms and nanomaterials [13]. Since then, the field of nanotechnology and nanoscience rapidly became a booming field of research, to which a vast majority of worldwide research agencies paid attention to, especially for industrial landscapes, due to the potential superior properties of nanoparticles and the possibility to create tailored products which overcome the limitations of current products and applications. To date, nanotechnology has countless applications in different fields and has already impacted our society and daily lives. For example, in health and biomedical areas, nanocarriers are currently used for targeted, triggered, and controlled delivery of drugs or other therapeutic molecules to specific areas of the body [14]; or silver nanoparticles are incorporated into commercial bandages to kill harmful microbes [15]; or nanoparticles of TiO_2_ are used as photoprotection (i.e., sunscreen) to mitigate the damage of UVB radiation and UVA2 in our skin [16]. In other areas, such as computer science, the development of nanotransistors allows electronic devices to store larger amounts of data in smaller sizes [17]. In the case of built heritage conservation, as in other scientific fields, nanotechnology has also provided the practice during the last 20 years with numerous novel materials made of nanoparticles, as in the case of consolidant agents and/or protective coatings inducing structures with properties such as self-cleaning, hydrophobic, biocide, insulation, air-purifying or solar protection, with potentially superior properties to conventional products and methods.

However, why does size matter? This is because essentially, on the nanometer scale, materials exhibit unusual properties that bulk materials do not have, such as quantum size (i.e., size- and shape-dependent characteristics) and surface modification effects. When the particle size is on the nanoscale, properties such as melting point, fluorescence, electrical conductivity, magnetitic permeability, or chemical reactivity change as a function of the size of the particle [19,20,21,22]. For example, when one changes the size of a particle, it can change other properties as for example its color reflectance, as is clearly observed in this simple experiment designed by Michael Faraday (1971–1867, [23]) with the nanoparticles of gold (Figure 1b). That is because, in nanometer-scale particles, the arrangement of atoms transmits and reflects light differently, and gold can appear light red or blue, whereas silver can appear yellowish or amber colored and thus different to regular-sized particles [19]. Please note that interactions with other materials might present different emissions of light and therefore color, as in the case of the Lycurgus Cup [6]. Hence, the essential feature of nanomaterials is that their physical and chemical properties are size dependent and this allows the possibility to arrange the material properties not only by defining its chemical composition, but also by tailoring the size, shape, morphology, and surface of the nanostructures, and this is strictly related to how individual atoms, molecules, or smaller nanoparticles are assembled [19]. 

The origin of the size dependence of the properties of nanoparticles is based, essentially, on two fundamental effects: (i) the surface/volume ratio of a nanoparticle increases as its size decreases [19]; and (ii) the limited dimensions of the nanoparticles lead to spatial confinement and scaling effects that affect a variety of different other properties [17]. Further, the properties of nanomaterials are also highly determined by their surface modifications [21]. The surface area-to-volume ratio is the amount of surface area per unit volume of an object, or collection of objects, and is essentially inspired by the old mathematical Galileo’s Square-Cube Law [24], which states that as the shape of a material grows in size, its volume grows faster than its surface area (Figure 2a). On the other hand, if a cube is divided into sub-units, the volume will remain constant while its surface area significantly increases (Figure 2b). Therefore, since nanoscale materials have much larger surface areas than similar masses of larger-scale materials, they have a greater amount of the material (in relation to its volume) exposed to the exterior that can come into contact with surrounding materials. This is a key concept in nanoscience and engineering and significantly affects several properties such as diffusion and heat transfer by conduction [25], mechanical properties [26], or reactivity [19]. As a consequence of size reduction and the surface-to-volume ratio, spatial confinement and scaling effects are fundamental to understanding why nanoparticle properties are different compared to their bulk materials [27]. When the particle dimension is reduced, the number of surface atoms and how they are structured, are also different [28]. In general, as the nanoparticle decreases, the number of atoms is gradually reduced, while the fraction of atoms located at the surface increases [18,19,29,30], which is essentially the so-called scaling effect [31]. Thus, since atoms at the surface have a coordination shell and are significantly different from those in the interior of the particle, surface atoms have fewer neighbors and are less stabilized than bulk atoms [19]. As a result, surface atoms have higher surface energy, higher reactivity, and increased mobility. Consequently, as the size of the nanoparticle gradually decreases, the contribution of the surface atoms to the total free energy and the properties of the nanoparticles are progressively modified: melting and evaporation temperature decrease, and the reactivity, elasticity, and plasticity increase [18,22,30,31,32]. Moreover, this also increases the ability of the nanoparticles to form stable colloidal dispersions, which has additional important consequences, as in the case of calcium hydroxide nanoparticles (the so-called nanolimes), as it allows nanoparticles to be dispersible in solvents and provide more suitable applications in the field [33]. However, based on their high surface area and large surface energy, nanomaterials tend to aggregate and fuse when their surfaces are uncoated, sometimes losing their unique properties. For this reason, stabilizers or additives are usually added to modify the surface to increase the stability and maintain the specific properties of nanomaterials [31]. Therefore, beyond the issue of size, the properties of nanoparticles are not only dependent on their size and shape but are also determined by surface modifications [21], although it is not always clear how the latter really determine the properties of the nanoparticles and research is currently ongoing [19]. The challenge with stabilizers remains in how to control the type, number, and conformation on the nanoparticle surfaces when considering their target application.

In the field of built heritage conservation, the design and synthesis of precisely tailor-made nanostructures are still ongoing. During the last two decades, conservators and scientists have been developing novel treatments to overcome, when needed, the limitations of traditional products and methods such as consolidant agents, cleaning materials, and/or protective coatings. A wide range of products are available and currently in use for cleaning (e.g., ethylenediamine tetraacetic acid (EDTA), ammonium bicarbonate, latex or clay poultices), consolidation (e.g., lime or barium hydroxide, organic polymers such as acrylics or epoxies as well as alkoxysilanes) or surface coatings (e.g., water-repellent, anti-graffiti, self-cleaning and/or biocidal products). These traditional methods, as in the case of synthetic polymers or alkoxysilanes, often lack crucial physical, chemical and/or mechanical compatibility with the original substrate, or do not provide a long-term and durable performance [33,34,35,36,37,38,39,40], crucial for conservation interventions. Within this context, the development in nanoscience, due to the different properties of nanostructured materials, provided an opportunity to overcome some of the limitations of traditional methods, such as compatibility, while also creating new tailor-made products. The pioneering laboratory for developing nanostructured materials for cultural heritage conservation was the Center for Colloid and Surface Science (CSGI—University of Florence, Italy). In 2000 [41], Baglioni and co-workers, following the path suggested by Matijević in the field of colloid synthesis [42], initially developed Ca(OH)_2_ nanoparticles for the consolidation of mural paintings [43], and stone [44]. Subsequently, nanoparticles specifically designed for cleaning, water repellency, as well as antimicrobial and anti-graffiti treatments were developed by other research teams [45,46]. There are currently several approaches for producing nanomaterials that can be specifically designed for the conservation of cultural heritage, including mostly chemical and physical methods [33,47,48,49]. Chemical approaches involve the synthesis of nanomaterials through chemical reactions [19,20]. The most common method is sol-gel synthesis, which involves the hydrolysis and condensation of metal alkoxides in a solution [33]. This approach can produce various nanomaterials for built heritage conservation such as silica nanoparticles and metal oxide nanoparticles, including Ca(OH)_2_ [33] TiO_2_ [49,50] and ZnO nanoparticles [49,51]. On the contrary, physical approaches, involve the physical processing of materials to produce nanoparticles [19]. One such method is high-energy ball milling, which involves the mechanical milling of bulk materials to reduce their size to the nanoscale [20]. This approach can produce metal and metal oxide nanoparticles, including CeO_2_ nanoparticles [46,50,52]. However, this approach is less popular in the heritage field since this requires the use of large quantities of energy [46,49,50]. A comprehensive review of nanomaterials for built heritage conservation can be found elsewhere [33,48,49]. 

For built heritage purposes, both chemical and physical approaches can have advantages and disadvantages, and the choice of method depends on the specific properties and applications of the nanomaterials in structures [19,20]. For example, chemical methods often offer better control over the size, shape, and composition of the nanoparticles, while physical methods can produce nanoparticles in larger quantities with a narrow size distribution [53]. Thus, the choice of production approach for nanomaterials depends, in general, on other factors such as the specific properties of the materials needed for a particular application or the required quantity [20,53]. However, in practical cases, the reality is that in most cases, the selected synthesis approach often depends on the availability of resources and equipment of conservation scientists. The potential benefits of nanomaterials for the production of novel methods/products tailored for built heritage conservation are essentially based on: (i) the possibility to design conservation products highly compatible with the original substrate; (ii) the possibility to create a product with higher reactivity and increased mobility, which are important features when dealing with processes as curing time, carbonation, cleaning actions, anti-pollutant procedures, anti-microbial or photocatalytic capacity; (iii) since particles have reduced dimensions of approximately 1–100 nanometers, there are also no major limitations of size in order to penetrate deep into damaged materials; (iv) since particles can be dispersed in alcohol media, creating a nanofluid, which minimizes aggregation phenomena, it also allows the application of higher amounts of more reactive nanostructured materials, and the alcohol evaporates usually leaving no residue or undesired side-effects. The size of nanoparticles in those nanofluids is a critical factor that affects their stability, properties, and potential applications [54,55,56,57,58] and that determines their potential applications [55,56]. One reason why size matters in nanofluids is that it affects the stability and homogeneity of the fluid [54,57]. Smaller nanoparticles have a higher surface area to volume ratio, which makes them more reactive but more prone to agglomeration and settling, leading to instability and non-uniformity of the fluid [54,55,58,59]. In contrast, larger nanoparticles are more stable and produce a more homogenous fluid [56] and it can be more effective based for specific substrates with high number of pores with large pore size diameter (>10 µm) [60]. The size of nanoparticles in nanofluids also affects their thermal and optical properties, as observed in the case of Michael Faraday’s experiment, which is shown in Figure 1b [23]. In this case, smaller nanoparticles have a higher surface area to volume ratio, which enhances their ability to transfer heat, leading to the increased thermal conductivity of the fluid. In the specific case of nanolime, among the most commonly used nanofluids in the practice, this nanomaterial was developed to overcome the limitations of the traditional limewater treatment, which has been used over centuries to consolidate deteriorated wall paintings, limestone or plaster. However, the main constraint of the limewater technique is the low solubility of lime particles in water (~1.7 g/L), the slow reactivity of lime particles due to their low surface area, and the low penetration of Ca(OH)_2_ particles into the substrate [33,34]. The nanolime consolidating effect takes place by the same mechanism as for the limewater technique (i.e., their conversion into cementing CaCO_3_ upon the reaction of Ca(OH)_2_ with atmospheric CO_2_), but the smaller size of the lime particles (nanoscale) improves their performance. In this sense, the advantages of nanolime compared to limewater are: (i) nanolimes contain higher amounts of calcium hydroxide (up to 50 g/L); (ii) lime nanoparticles are more reactive due to their higher specific surface, thus increasing the carbonation rate; (iii) nanolimes can penetrate deeper into the substrate because of their smaller particle size; (iv) nanolimes have better colloidal stability due to their smaller particle size and the repulsive electrostatic forces when dispersed in short-chain aliphatic alcohols; or, (v) reduced whitening of the treated surface [33,34,59,60,61]. 

In the case of dispersion of Titanium dioxide (TiO_2_) nanoparticles, which is another highly used nanofluid, this product has also been extensively used in built heritage conservation due to their photocatalytic and self-cleaning properties, making it an attractive option for the treatment of surfaces which are prone to deteriorate [49,50,53]. TiO_2_ nanoparticles can be used to create thin films on the surface of building materials, which can protect them from UV radiation, pollutants, and biological growth, while also facilitating self-cleaning through photocatalysis [49]. In addition, TiO_2_ nanoparticles have also shown potential enhancement of mechanical strength and resistance to weathering [50]. In this case, the small size of TiO_2_ nanoparticles also allows for deeper penetration into the substrate, which can improve their effectiveness by providing higher adhesion, enhancement of mechanical properties and increasing its resistance to weathering [52], also protecting the underlying material [53]. TiO_2_ nanoparticles in the nanoscale range have a larger surface area-to-volume ratio than larger particles, which enhances their photocatalytic and self-cleaning properties [50] while also allowing greater interaction with environmental pollutants and organic matter [49]. 

In the case of silica (SiO_2_) nanoparticles, another common nanofluid used in built heritage conservation, these silica nanoparticles are used in the consolidation of degraded stone, such as limestone, to improve its mechanical strength and prevent further degradation [53]. As in the case of Ca(OH)_2_ and TiO_2_ nanoparticles, the reduced size of silica nanoparticles allows particles to penetrate deeply into the substrate, enhancing their consolidation and strengthening effects [62]. Furthermore, also the high specific surface area of silica nanoparticles in the nanoscale range allows for greater interaction with the substrate and bonds, leading to improved adhesion and consolidation [63]. Silica nanoparticles have also been shown to improve the water-resistance of treated surfaces and provide protection against environmental pollutants and organic matter [62,63], while also providing a higher reduction in the formation of cracks during the formation of the silica gel [49]. However, it should be noted that the size and shape of silica nanoparticles can affect their behavior and interactions with the substrate, especially for calcareous substrates [64], and thus careful further considerations of the synthesis and application methods are necessary for the successful use of silica nanoparticles in conservation. 

Zinc oxide (ZnO) nanoparticles have also been widely used for the conservation of built heritage materials due to their excellent UV-blocking properties which are able to absorb a wide range of UV radiation, thus protecting the underlying materials from photodegradation and discoloration caused by exposure to sunlight [51,64,65,66,67]. Additionally, ZnO nanoparticles have shown potential in the consolidation of building materials such as stone, brick, and mortar, improving their mechanical strength and durability [65]. As in previous nanofluids, the reduced size of ZnO nanoparticles also allows higher reactivity and protection [51,66] and deeper penetration into the substrate, resulting in better adhesion and enhanced strength and durability of the material [67]. Furthermore, since ZnO nanoparticles also exhibit antibacterial properties, being this useful in the prevention of microbial growth and biofouling on building surfaces [66], which has been especially useful in humid environments [53]. 

Another example is the use of cerium oxide (CeO_2_) nanoparticles for treating corrosion on stone or metal surfaces, such as bronze and copper alloys [46,49,50,52]. In this case of metals, the reduced size of CeO_2_ nanoparticles allows for their penetration into the porous surface, where react with the corrosion products and neutralize them through redox reactions [46]. However, in the case of stone treatments, CeO_2_ nanoparticles have been used in combination with other products to also induce corrosion resistance due to their high surface area, high reactivity, and redox activity [52]. Similar to previous examples, the smaller size and associated larger surface area-to-volume ratio of CeO_2_ nanoparticles enhance their reactivity and catalytic activity [46], allowing for more effective removal of pollutants from treated surfaces [52]. The use of CeO_2_ nanoparticles in these coatings has been also shown to enhance the mechanical properties of the treated surfaces, such as compressive strength and elasticity, thereby improving their resistance to weathering [52]. While CeO_2_ nanoparticles show significant potential for the conservation of built heritage materials, further research is still needed to fully understand their effectiveness, applications, and long-term durability.

## Conclusions

During the first period of the 21st century, the field of built heritage conservation has witnessed the flourishing of new nanomaterials for conservation with significant in situ practice applications. Overall, these nanomaterial products could offer promising solutions for the conservation of built heritage, providing improved performance and sustainability compared to traditional conservation materials. However, several future challenges remain to be solved. The questions now are, at this point, should the emphasis of heritage conservation be placed on the development of new nanomaterials and their application procedures? Are most of the nanomaterials already in use precisely understood and specifically tailored to optimize their effectiveness during application? Are there any significant data about testing the effectiveness of nanomaterials that are reproducible in both laboratory and practice activities? How can we integrate the concepts of sustainability to create greener materials to solve environmental problems? Are the techniques and methods for evaluating nanomaterials and their effectiveness accessible to conservation practitioners? Does science need to provide more research to evaluate the long-term durability of nanomaterial treatments? In this article, we have discussed the developments in nanoparticles and some potential applications in the built heritage conservation sector. Insofar as nanoscience is concerned, we already have areas, such as nanolime and nanoparticles of TiO_2_, which are witnessing rapid development and promising results. However, there are other fields where there has been significantly less progress. There is no doubt that nanoscience opens a new door to new applications and products that can potentially revolutionize the practice field and how we conserve built heritage materials. However, we are living at the beginning of this era and this idea of developing products tailored to specific needs still needs more time and further research. Further, we consider that this new direction should embrace the concept of green chemistry and sustainability to contribute to global environmental problems, fully aligned with Agenda 2030.

## Figures and Tables

**Figure 1 materials-16-03277-f001:**
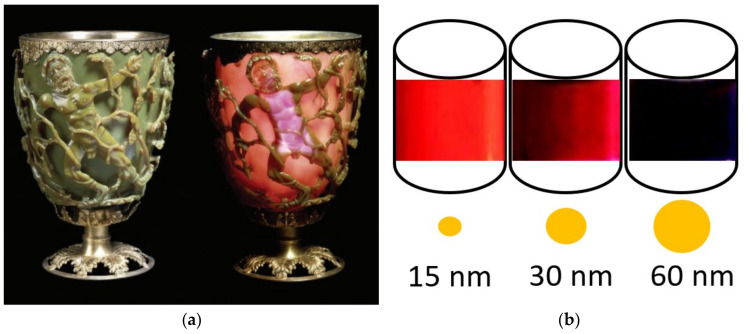
(**a**) The Lycurgus Cup (British Museum; AD fourth century, Room 41, i.d. 1958,1202.1). This Roman cup is made of ruby glass. When viewed in reflected light, for example in daylight, it appears green (**left**). However, when a light is shone into the cup and transmitted through the glass, it appears red (**right**). Images reproduced, with permission from [5]. © The Trustees of the British Museum. Image shared under a Creative Commons Attribution-Non Commercial-ShareAlike 4.0 International (CC BY-NC-SA 4.0) license. (**b**) Colloidal suspensions of gold nanoparticles in water. The nanoparticle diameters are indicated on the container (Image adapted from Celso de Mello Donegá, Nanoparticles workhorses of Nanoscience, Springer, 2014, courtesy F.H. Reincke, Utrecht University [18]).

**Figure 2 materials-16-03277-f002:**
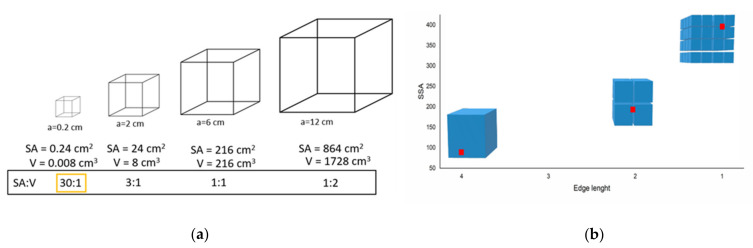
(**a**) Size dependence of the surface-to-volume ratio for hypothetical cubic samples. The SA:V (surface area-to-volume) ratio is represented in the rectangle to show how this ratio increases when particle size is decreased; (**b**) the model of the evolution of specific surface area as compared to the edge length. Red squares represent the data obtained from Baglioni et al. [34] and blue cubes represent the model of the particles at constant volume.

## Data Availability

Not applicable no new data.

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
