# Peer review of "The Boom in Nanomaterials for Built Heritage Conservation: Why Does Size Matter?"

_materials, 2023, doi:10.3390/ma16083277_

Round 1

Reviewer 1 Report

This is one of the best articles that I have reviewed. It is very interesting for the readers specially for those new to the field. I congrats the authors for their excellent work. I actually wanted to keep on reading. I believe that the article could have been even better if  the authors included the different nanomaterials production approaches (i.e., chemical and physical approaches). In addition, I hoped to read some information regarding 'Nanofluids'. When, why, and how are they produced and what applications can they be used in today.

I think the authors will highly benefit from including such details in their proposed manuscript. Other then that, I highly support the publishing of the in-hand article after taking into consideration the previous recommendations and the kind approval of the respected Editor.

Well done and all the best,   

Author Response

Please kindly see the attachment the answers

Reviewer 2 Report

This is a nice manuscript of mini review – opinion article about nanomaterials used in build cultural heritage; rather informative and pleasurable to read. However, please consider that the article does not follow the common structure of review papers, for example the is no Introduction section. You need to add subtitles in the text to structure it better and to make it easier to follow for the potential readers; especially since this is not a classical review paper and thus you need to better target your goals and your audience. Furthermore, I suppose that the paper needs to set more evident scientific goals (now they are rather vague and general) on the issue that it is trying to address (nanomaterials in built cultural heritage), in order to have an impact in the literature.

Please add in the abstract shortly the main concept of your arguments in the attempt of answering the main questions set. Also, add shortly the answer of the questions that you are trying to address.

 Lines 31, 37, 64 : Please try to replace the word “Manipulation/ manipulate”. It is evident what you are trying to say; however, it would be better to use another word (with no negative meanings), which it will reflect better the notion of “handling” managing (?) nano-matter.

After line 82, you need to add and/or analyze more about the kind of nano-products currently used and/or developing for the field of cultural heritage. There is an imbalance here, since you do not analyze sufficiently the uses of nano materials in the field of cultural heritage.

You are only analyzing the nanolime materials, while in the title, abstract etc you are in general referring to nanomaterials. Please consider either to change the title/abstract or to add at least one more major nanomaterial currently in use in the conservation field (maybe TiO2 particles and their applications???)

Also, it would be a good idea to add the concept and the applications of some commercial nanoproducts, as well as to further analyze the impact and the concept of green materials and how you consider that nanomaterials can be produced in terms of sustainability following green chemistry concept.  

Author Response

(The authors gave the same response as above.)

Reviewer 3 Report

Although in its first part the article reports some interesting and unusual examples of nanomaterials and nanotechnologies used in the distant past to modify the properties of matter and objects of artistic value, most of the text focuses on describing the aspects that regulate the dependence of the properties of materials with their size, with a series of references to various application sectors, of which only some concern Cultural Heritage and very few concern built heritage, that is supposed to be the main topic.

Only at page 5 the article finally deals with nanomaterials for the conservation of built heritage, mainly focussing on one type, nanolime.

The potential benefits of nanomaterials for the production of new methods/tailored products for the conservation of built heritage have been known for about twenty years and very comprehensive reviews have already been published on the subject (some cited in the bibliography). Therefore I would have expected different aspects to be explored and debated in this opinion paper/review, some of which are only hastily mentioned in the conclusions: the fact that apart from nanolime and TiO2 nanoparticles the other nanomaterials for built heritage are struggling to establish themselves, which ones are exactly these other materials, how nanomaterials or nanotechnologies applied to built heritage or conservation in general can play a role in environmental sustainability.

For these reasons I believe that the article needs major revisions that deepen these themes, still little discussed and not already included in other reviews.

Author Response

(The authors gave the same response as above.)

Reviewer 4 Report

Dear Authors,

I appreciate that this short review opens an exciting discussion chapter about the main criteria nanomaterials should obey In cultural heritage.

Is an excellent idea, and personally I support this proposal, to analyze more focused the nanomaterials and nanosciences in this area.

If I will be able to contribute somehow, I will do it.

Congratulations!

1. What is the main question addressed by the research?  The paper is an editorial, very well focused on the nano subject and their applications on CH conservation and restoration.   2. Do you consider the topic original or relevant in the field? Does it  address a specific gap in the field?  Original paper, it address a specific gap in the field
3. What does it add to the subject area compared with other published  material?
The editorial is based on the authors expertise and some representative papers could be identified at these authors.

4. What specific improvements should the authors consider regarding the 
methodology? What further controls should be considered? 

The paper is extrmely important, is an open editorial, pointing out the size of nano and their efficacity in conservation and restoration.  

5. Are the conclusions consistent with the evidence and arguments presented 
and do they address the main question posed? YES

6. Are the references appropriate? YES

Author Response

(The authors gave the same response as above.)

Round 2

Reviewer 2 Report

The manuscript can be accepted in its current form.

Reviewer 3 Report

The revised version has been greatly improved and most of the comments made by the reviewers have been answered so I consider the article now fit for publication